# A Personalized Approach to Maintaining Brain Drainage: A Case Series with a Technical Note

**DOI:** 10.3390/jpm15070264

**Published:** 2025-06-20

**Authors:** Manuel Moneti, Anna Malfatto, Ernesto Migliorino, Antonio Bassoli, Mariangela Chiarito, Claudia Iulianella, Noemi Miglionico, Luca Bombarda, Carlo Alberto Castioni, Carlo Bortolotti, Antonino Scibilia, Corrado Zenesini, Raffaele Aspide

**Affiliations:** 1Anesthesia and Neurointensive Care Unit, IRCCS Istituto delle Scienze Neurologiche di Bologna, 40139 Bologna, Italy; manuel.moneti@ausl.bologna.it (M.M.); anna.malfatto@ausl.bologna.it (A.M.); carloalberto.castioni@ausl.bologna.it (C.A.C.); raffaele.aspide@isnb.it (R.A.); 2Azienda Unità Sanitaria Locale Di Bologna, Intensive and Critical Area Unit, 40139 Bologna, Italy; antonio.bassoli@ausl.bologna.it; 3School of Anesthesia and Intensive Care, University of Bologna, 40139 Bologna, Italy; mariangela.chiarito@studio.unibo.it (M.C.); claudia.iulianella@studio.unibo.it (C.I.); noemi.miglionico@studio.unibo.it (N.M.); 4School of Emergency Medicine, University of Bologna, 40139 Bologna, Italy; luca.bombarda@studio.unibo.it; 5Neurosurgery Unit, IRCCS Istituto delle Scienze Neurologiche di Bologna, 40139 Bologna, Italy; carlo.bortolotti@isnb.it (C.B.); antonino.scibilia@ausl.bologna.it (A.S.); 6 Epidemiology and Statistic Unit, IRCCS Istituto delle Scienze Neurologiche di Bologna, 40139 Bologna, Italy; corrado.zenesini@ausl.bologna.it

**Keywords:** urokinase, fibrinolysis, external ventricular drain, CLEAR, speed clot resolution

## Abstract

**Background/Objectives**: The percutaneous insertion of an external ventricular drain (EVD) is a common neurosurgical procedure that is crucial in managing acute brain injuries because of the drain’s role in monitoring intracranial pressure and draining cerebrospinal fluid. The primary indication is acute hydrocephalus, which often results from subarachnoid hemorrhage, intracranial hemorrhage, traumatic brain injury, stroke, or infection. Standard EVD placement targets the frontal horn of the lateral ventricle. However, complications such as hemorrhage, infection, and catheter occlusion frequently arise, with occlusion rates ranging from 19% to 47%. Occlusion can lead to increased intracranial pressure, necessitating interventions such as saline flushes or fibrinolytic drug administration. The placement of an EVD is a very specific choice that must be tailored to the individual patient, often in scenarios in which multiple interpretations of the data are possible: the question of which patient is eligible for EVD placement may be subjective. Intraventricular fibrinolysis (IVF) with urokinase-type plasminogen activator (uPA) or tissue-type plasminogen activator is used with the aim of lysing intraventricular clots and preventing EVD occlusion. Despite numerous studies, conclusive evidence on their efficacy is lacking. The CLEAR III trial confirmed the safety of IVF but showed uncertain benefits in neurological outcomes. Given the limited literature on uPA, this study evaluates its intrathecal administration for the prevention of EVD occlusion. Not all therapies are appropriate for all patients, and customizing strategies is often the right way to get the best result. **Methods**: This retrospective study analyzed 20 patients with EVDs receiving intrathecal uPA. The patients had a mean age of 56.4 years, with 95% presenting with hydrocephalus and 80% presenting with intraventricular hemorrhage. uPA dosages varied (25,000–100,000 IU), with an average of 3.9 doses per patient. **Results**: IVF effectively maintained EVD patency in 95% of cases. One patient experienced asymptomatic bleeding, while four (20%) developed post-treatment infections, the development of which was potentially influenced by the prolonged duration of EVD retention (>21 days). Analysis of Graeb scores showed faster clot resolution with early uPA administration. A higher initial Graeb score correlated with increased total uPA load but not with mortality or discharge outcomes. Although infection rates were slightly higher than in CLEAR III, multiple confounding factors, including duration of EVD retention and bilateral placement, were present. **Conclusions**: This study supports the feasibility and safety of intrathecal uPA administration for management of EVD occlusion in certain contexts. The appropriate choice in the context of ‘personalized medicine’ must necessarily consider the risk–benefit ratio.

## 1. Introduction

The percutaneous insertion of an external ventricular drain (EVD) is one of the most common procedures performed in neurosurgery due to its usefulness in many acute brain injuries (ABIs). Its safe execution may allow the measurement of intracranial pressure (ICP) and the drainage of cerebrospinal fluid (CSF) [1]. The main indication for EVD placement is acute hydrocephalus, which can be an early complication of many ABIs such as subarachnoid hemorrhage (SAH), intracranial hemorrhage (ICH), traumatic brain injury (TBI), infections, acute ischemic stroke, and failure of a previously inserted device, such as an EVD, ventriculoperitoneal shunt (VPS) or ventriculo-atrial shunt (VAS) [2,3,4]. However, in a given population of neurosurgeons managing ABI patients, there will be multiple attitudes towards treatment, with different clinicians being more or less aggressive and interventional; this therapeutic choice is marked by a certain degree of heterogeneity.

The most common EVD-insertion technique involves inserting the catheter in the frontal region starting from Kocher’s point, targeting its tip to the frontal horn of the lateral ventricle at the level of Monro’s foramen. Once correct placement has been confirmed, the catheter can be connected to a drainage system. Even though EVD insertion is often seen as a minor surgical procedure, it can be associated with various complications, most commonly hemorrhage, infections, and brain damage secondary to catheter malfunction, most often after occlusion [5]. Published data report an all-causes occlusion rate between 19% and 47%, and occlusion is associated with numerous complications and additional costs [6,7,8]. EVD lumen occlusion can be secondary to adhesion and invasion by cerebral tissue in VPSs and VASs but is most often secondary to blood clots and the accumulation of cellular debris in EVDs [5,9,10]. Furthermore, dislocation of the EVD tip into the cerebral parenchyma can be often secondary to cerebral edema and midline shift resolution [11]. The predictors of EVD occlusion are not known; however, large amounts of intraventricular blood are certainly a possible trigger for occlusion.

Whatever the mechanism, the consequence of EVD occlusion is the immediate interruption of CSF drainage, which, in patients with reduced brain compliance, can result in increased ICP. When the CSF drainage drops below a certain volume and the ICP waveforms indicate altered fluid coupling, there are several interventions that can be tried. The administration of small amounts of sterile crystalloid solution can dislodge clots that occlude the catheter holes [12]. This procedure can cause an increase in ICP and has been associated with various complications, including pneumocephalus, peri-catheter cerebral edema, and, most significantly, new-onset cerebral hemorrhage [10]. Alternatively, some centers administer fibrinolytic drugs through the catheter to aid in the lysis of intra- and peri-catheter clots [13,14,15]. This drug administration should be preceded by a computer tomography (CT) scan to identify any malpositioning of the catheter (intraparenchymal position is a contraindication to fibrinolytic treatment) and reveal other possible causes of catheter malfunction [16]. The choice of which patient or EVD should receive intervention is not easy; everything has to be put in the context of ‘personalized medicine’.

If no treatment is effective, the catheter must be removed and a new one must be placed. The replacement of an occluded EVD is associated with an increase in the rate of hemorrhagic complications (as high as 41% of cases [17]). Many of these new-onset hemorrhages are asymptomatic, but some of them can be large enough to cause neurologic alterations and even increase mortality rates. Every EVD replacement increases the risk of infective complications, as has been demonstrated in many studies [7,18,19]; this risk increases with multiple replacements [20], even when antibiotic-impregnated catheters are used [19].

Intraventricular fibrinolysis (IVF) is the administration of a thrombolytic drug through an EVD. Two different drugs can be used: urokinase-type plasminogen activator (uPA) and tissue-type plasminogen activator (r-tPA). The objectives of these treatments are the early lysis of intraventricular clots and the avoidance of drainage occlusion. Between 1990 and 2000, more than 200 studies about fibrinolytic treatment were published, but no conclusive evidence on its efficacy has emerged, even though some case series of patients treated with uPA showed better outcomes when these patients were compared with historical controls [21,22]. A Cochrane review published in the early 2000s collected data from seven independent studies, for a total of 74 patients with intraventricular hemorrhage (IVH) associated with ICH or SAH (17 treated with uPA and 57 with rTPA). The results showed good neurological outcomes for 50 out of 74 patients and limited findings of complications, as follows: bacterial meningitis (5 patients), increase in hematoma ( 1 patient), extradural bleeding ( 2 patients). The cornerstone study is the CLEAR III study “Clot Lysis Evaluation of Accelerated Resolution of Intraventricular Hemorrhage” [23], which is cited in the AHA/ASA [10] guidelines for the management of ICH. Based on that study, clot lysis is considered safe for patients with a Glasgow Coma Scale (GCS) score of 3 who have IVH primary or secondary to ICH with a volume >30 mL and who require placement of an EVD due to obstructive hydrocephalus, as this intervention reduces mortality compared to EVD placement alone, albeit with uncertain benefits for outcome (evidence class 2b). Despite that finding, the literature lacks data on dosages, timing of administration, precautions for use, and the use of uPA. In 1999, the production and distribution of uPA in the United States was suspended by the Food and Drug Administration due to reports of several adverse events; the drug was then reintroduced to the market and is most widely used as an intra-arterial thrombolytic agent for acute ischemic stroke. It is a low-cost drug.

The aim of the present study is to investigate the safety of intrathecal administration of uPA through an EVD as part of a personalized approach to resolving impending catheter occlusion. This study describes the administration procedure and the incidence of treatment-related complications (infections and hemorrhage) in a single-center series of patients. The basic purpose of the study is to support clinicians in the critical area of making the right choice for the right patient, guided by principles of safety.

## 2. Materials and Methods

### 2.1. Study Design, Participants and Setting

This observational retrospective monocentric study was conducted between May and December 2023. Patients 18 years of age or older with a diagnosis of ABI who had been admitted to the intensive care unit (ICU) of the IRCCS Institute of Neurological Sciences of Bologna and treated with intrathecal uPA via an existing EVD were considered eligible for recruitment. Only patients with subarachnoid hemorrhage and intraventricular hemorrhage who had a large amount of blood in the EVD and a secured aneurysm who either were at risk of drainage-catheter occlusion due to clot formation or had recently experienced malfunction of the EVD were administered uPA; uPA was not administered to patients without a secured aneurysm or patients with a normally functioning EVD. Importantly, in our center, uPA, in cases of SAH, is administered only after the securitization of the aneurysm (via coiling or clipping). The data were extracted anonymously from the computerized files in use in the ICU. The study was conducted in accordance with the Declaration of Helsinki and the general principles of good clinical practice. The Institutional Review Board approved the study (Cod. CE22203). The authors followed the Strengthening the Reporting of Observational Studies in Epidemiology (STROBE) guidelines for cohort studies (http://www.strobe-statement.org (accessed on 20 February 2023)) [24].

### 2.2. Demographic and Clinical Variables

The age and sex of all patients were collected as demographic variables. Weight, height, body mass index (BMI), comorbidities, diagnosis on entry to the ICU, onset GCS score (range from 3 to 15), World Federation Neurological Surgeon (WFNS, SAH only—range from 1 [no deficit] to 5) modified Fisher Scale score (SAH only—range from 1 [no SAH] to 4), timing of first CT pre-EVD, number and timing of post-uPA CT scans, modified Graeb score [25] for each CT (estimate of the amount of blood in the ventricles, range from 0 to 32), length of stay (LOS) in the ICU, presence of IVH and/or hydrocephalus, ICP > 20 mmHg, EVD position (left/right), placement of dual EVD, number of administrations of uPA, days with the EVD in place, EVD replacement, days of uPA administration, total uPA load (the total amount of uPA administered to the individual patient), and the use of low-molecular-weight heparin (LMWH) and/or anticoagulant and/or antiplatelet therapy concurrent with uPA administration were recorded. The modified Graeb score is a useful tool with which to quantify the blood in the cerebral ventricles in IVH; it is calculated by assigning a score to each ventricle based on the amount of blood in each compartment. The compartments considered are as follows: lateral ventricles (right and left, maximum score 4 each), temporal horns of the lateral ventricles (right and left, maximum score 2 each), occipital horns of the lateral ventricles (right and left, maximum score 2 each), third ventricle (maximum score 4), fourth ventricle (maximum score 4); an additional point is added for each compartment if it is expanded because of the clot.

### 2.3. Outcomes

Bleeding, infection/meningitis/ventriculitis, timing of speed clot resolutionVPS or VAS positioning, ICU death, Glasgow Outcome Scale Extended (GOS-E) score at ICU discharge were the outcomes considered. .

### 2.4. Statistical Analysis

In the descriptive analysis, continuous variables are presented as the mean standard deviation (SD) or median and interquartile range (IQR), while categorical variables are presented as absolute (n) and relative (%) frequencies.

A linear mixed-effect modelling analysis with random slope and intercept was applied to evaluate the longitudinal change in the Graeb Score (dependent variable) over time (independent variable, in hours). The overall error distribution in linear mixed-effects models is assumed to be Gaussian. Additionally, the model can accommodate heteroskedasticity and within-group correlation structures by specifying appropriate covariance structures for the random effects and residuals. This flexibility allows for accurate modeling of longitudinal data in which repeated measures are nested within subjects. The results are presented as β coefficients with 95% confidence intervals (95% CIs) that represent the range within which the true population parameter is expected to lie with 95% confidence, assuming the model is correctly specified and its assumptions are met. The results of the main analysis were stratified for the following variables: time to uPA administration and number of drainages to detect if the slope was different between subgroups.

The association between categorical and continuous variables, such as EVD duration, GOS-E at discharge, and total uPA load, were evaluated by the Kruskal–Wallis test. The correlation between continuous variables were evaluated by Spearman’s Rho coefficient.

Statistical analysis was performed using Stata statistical software version 14.2 (StataCorp LLC, College Station, TX, USA).

## 3. Results

A total of 20 patients were included in this work; 10 (50%) were female, and the patients’ ages ranged between 25 and 79 years (mean age 56.4 SD ± 15.5 years). The anthropometric variables examined, diagnosis at admission to ICU, comorbidities, GCS, and the presence of IVH and/or hydrocephalus were as described in Table 1.

At onset, the patients presented a median GCS score of 7.9 ± 4.0; 95% presented hydrocephalus, and IVH was present in 80% of cases.

All patients with SAH at onset (n = 7) had a modified Fisher score of 4 (worst score); 57% of SAH patients had a WFNS of 5, while 29% had a WFNS of 4 (severe condition) and the remaining 14% had a WFNS of 2. All 20 patients underwent EVD placement, with 14 (70%) undergoing placement on the right side, while 6 (30%) underwent placement bilaterally, sometimes in later stages.

The average time between EVD placement and the first IVF dose was 2.8 days (SD ± 2 days). The first CT scan control was performed on average 60.6 h after the first uPA administration. uPA was administered in three different dosages: 25,000 IU, 50,000 IU, or 100,000 IU. Of the 77 total doses, 18 (23%) were 25,000 IU, 58 (75%) were 50,000 IU, and 1 (1%) was 100,000 IU. Five patients received different dosages during treatment, of whom four received doses of 25,000 IU and 50,000 IU, while one received a dose of 50,000 IU and a dose of 100,000 IU.

Overall, an average of 3.9 intraventricular administrations of uPA were carried out for each patient. However, some subgroups of patients are highlighted on the basis of the total doses given. In particular, we identified a cluster of patients treated with one to two doses (n = 12, 60%), a subgroup treated with four to five doses (n = 4, 20%), and a subgroup treated with at least eight total doses (n = 4, 20%). The average duration of treatment was 3 days. Patients received a single administration per day or two administrations in 24 h.

During the administration of IVF, 16 patients (80%) were undergoing pharmacological LMWH thromboprophylaxis; 4 of these (15% of all patients) received LMWH at an anticoagulant dosage.

The total average duration of EVD treatment was 21.4 ± 13.4 days. In 95% of cases, treatment with intraventricular uPA proved effective in maintaining patency or restoring the patency of the EVD following sub-occlusion. In only one case did the catheter obstruction prove to be refractory to treatment, requiring catheter replacement. A second patient needed catheter replacement after treatment with intraventricular uPA, but in that case, replacement was needed for infection control.

In one case (5%), asymptomatic bleeding occurred following IVF. This complication occurred after the second dose (50,000 UI) of uPA and did not involve alterations in the state of consciousness, and total intraventricular blood volume increased slightly at subsequent checks (total number of doses: eight).

Eight of the patients observed (40%) received a diagnosis of central nervous system (CNS) infection during hospitalization in an intensive care setting (clinical signs and laboratory confirmation with positive culture tests on CSF). Of these, four had of CNS infection that preceded EVD placement or uPA administration, while four (20% of all the patients treated) were diagnosed following treatment with uPA. In two cases, more than 7 days passed between the administration of uPA and the diagnosis of infection/start of antibiotic therapy, while in the other two cases, only two days passed.

All enrolled patients had undergone CT scans within 24 h prior to uPA administration. The Graeb score was recorded for 19 out of 20 patients. The mean value before uPA treatment was 12.9 SD +/− 10.8.

A total of 5 out of 20 patients (25%) underwent VPS placement; in half of the cases, this occurred 4–5 months after the hemorrhagic episode due to late recurrence of chronic hydrocephalus. The average length of stay in intensive care was 23.2 days ± 16.9 days. There were six (30%) deaths in the ICU. The mean GOS-E value at ICU discharge, including values for deceased patients, was 3.8.

## 4. Speed of Clot Resolution and Correlations

Correlation analysis showed that the Graeb score at onset negatively correlated with the initial GCS of the patients (Rho = −0.493). Although the correlation was not statistically significant, the data show that deceased patients in the enrolled cohort had much higher Graeb score values than ICU survivors. Furthermore, the Graeb score value at onset correlated positively with the total uPA load.

A negative trend was observed in the individual Graeb scores over time (see Figure 1), with a longitudinal coefficient (β) of −0.023 (95% CI −0.030 to −0.016). This coefficient represents the “speed of clot resolution” and indicates that for every additional hour, the Graeb score decreased by 0.023 points (95% CI −0.030 to −0.016), with a statistically significant *p*-value of <0.001. The stratification of β over time shows that in patients in whom uPA was administered early, ventricular clearance was faster (steeper slope): when uPA was administered in days 0–1, the value of β was −0.040 (95% CI −0.054; −0.026), *p* < 0.001. Furthermore, the β coefficient was higher in patients in whom the ventricular drain had remained in place longer and in patients in whom two drains had been placed: when the number of days with the EVD in place was >27, the value of β was −0.031 (95% CI −0.044; −0.019), <0.001.

Furthermore, the total amount of uPA administered correlated positively with LOS (Rho = 0.527). However, on application of the Spearman test, it was found that ICP (0.399), LOS (0.527) and initial Graeb score (0.574) correlated with total uPA load. However, no correlation was found between the total amount of uPA administered and the GOS-E at ICU discharge, although there was a minimal negative correlation between Graeb score at presentation and GOS-E at ICU discharge.

Furthermore, a correlation was identified between the duration of EVD treatment and the incidence of infections/ventriculitis (*p* = 0.032; 18.2 days vs. 32.0 days). GOS-E at discharge also correlated with infections/ventriculitis (*p* = 0.050; 3.1 vs. 5.6). No association was found between the incidence of infections/ventriculitis and total uPA load. As would be expected, patients who developed infections had a greater total number of days with the EVD in place.

## 5. Discussion

This study reports on a case series of 20 patients admitted to a neuro-ICU who underwent EVD placement. The clinicians, in a ‘customized approach’, chose to treat with intraventricular administration of uPA for impending catheter occlusion. Patients were identified as eligible for this treatment on the basis of multiple factors: the presence of a large amount of blood in the EVD and in the cerebral ventricles, creating a risk for EVD occlusion; and an initial malfunctioning of the EVD, as manifested in a reduction of CSF supply, after verification of the correct positioning of the EVD. If the procedure is performed in aneurysmal SAH patients, the uPA is always administered after the securitization of the aneurysm. The present case series demonstrates the safety and feasibility of employing this treatment to avoid complete catheter occlusion, a complication that often occurs, especially but not only, in the presence of large quantities of blood (mean value of first Graeb score 12.9). The clinicians’ choices took into account a number of elements, including the patient’s ongoing need for the EVD and the safety conditions for performing IVF (see technical notes in the supplemental materials). The customized strategy paid off: there was only one case of bleeding related to uPA administration. However, cases of infection appear to have been more frequent, occurring in 20% of the patients. Nevertheless, several infectious risk factors unrelated to fibrinolytic administration were identified: presence of bilateral EVD in three out of four cases, replacement of EVD due to dislocation in one case, attempt to restore catheter patency by irrigation with saline solution in two out of four cases, and a duration of EVD treatment before diagnosis of infection >21 days in two out of four cases; the maximum duration of EVD treatment is still debated in the scientific literature, and further studies should aim to identify a potential cut-off number for the duration of EVD treatment in days. One patient received a diagnosis of and treatment for suspected CNS infection during hospitalization in the neurosurgery department without a microbiological agent being isolated. This patent was administered two doses of uPA, and the suspicion of infection arose 7 days after the first administration of IVF. The duration of EVD treatment before the suspicion of infection was 8 days. Finally, the β coefficient interestingly indicates that IVF is effective, especially if it is administered early.

From comparison with data available in the literature, the CLEAR-III trial demonstrated the safety of applying a rigid therapeutic protocol associated with strict control of infectious and hemorrhagic risk factors (clot stability, coagulopathy, exclusion of vascular anomalies) [23]. The rate of infection of EVD falls within the range 0–40% [21]. In the CLEAR-III trial, this rate was indeed very low (4%), and those findings had a significant influence on some subsequent reviews and metanalyses [26,27,28] including the more recent metanalysis by Haldrup 2023 [29], who demonstrated a reduction in the incidence of infection from 12.8% of the group given EVD without IVF to 8.1% of the group given EVD with IVF. This protective effect against ventriculitis could be attributed to faster ventricular clearance, with an accompanying reduction in the duration of the EVD treatment itself, which also significantly reduces the risk of occlusion (from 37.3% to 10.6%) [29]. However, conflicting results emerge from the literature, and another meta-analysis reported increased risk of infection with IVF [30]. The retrospective review of Krel [31] demonstrated a ventriculitis rate of 18%, with a significant effect on ICU LOS. There is a great range of variability in the incidences of infection related to IVF reported in the scientific literature; the incidence of infection found in our study is higher than that observed in the CLEAR-III trial but aligns with the range observed in the literature; these findings suggest a need for larger prospective trials that compare IVF protocols while accounting for the duration of EVD treatment, which in our cases was longer.

The CLEAR III trial shows a rate of new bleeding of 16.8%, with symptomatic hemorrhages in 2.4% of cases. Recent meta-analyses predict an increased but not significantly increased risk (Solinge 2020 [27]; 8.6% vs. 6% in Kuramatsu 2022 [28]), but these results are affected by the high heterogeneity of the studies. In conclusion, the literature confirms the safety profile of IVF.

In light of the results obtained at a center that has begun using this type of procedure for prophylactic purposes, we feel we should list some key points. These are informed by the limited guidance available in the literature, multidisciplinary discussions, and bedside clinical practice [32], and are intended to guide physicians in selecting the appropriate patient and EVD. The Appendix A include our proposed treatment bundle, which outlines indications for treatment, dosage adjustments, and safe administration procedures. We open with “GHOW!” as a take-home-message (Figure 2).

**Figure 2 jpm-15-00264-f002:**
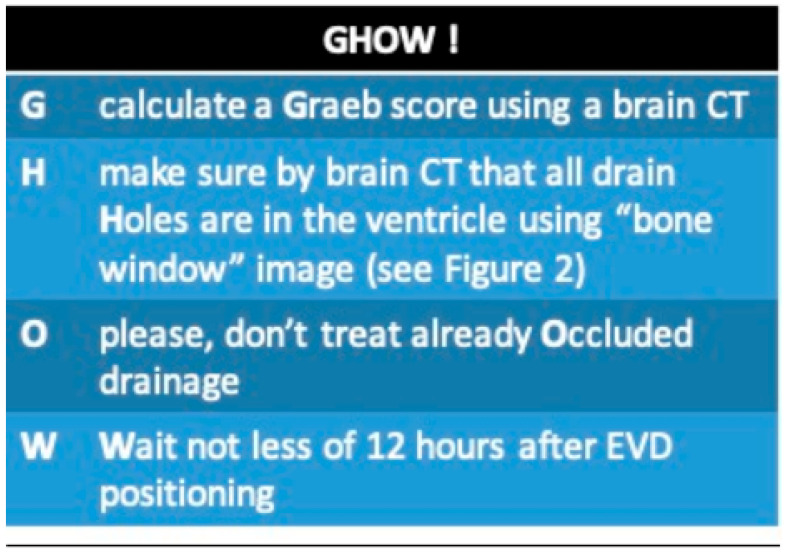
GHOW: a practical bundle for approaching intrathecal fibrinolysis.

## 6. Conclusions

EVD placement is very often a life-saving treatment. Maintaining EVD patency is essential for continuous CSF drainage and ICP monitoring. Repositioning an EVD due to occlusion is associated with a significant increase in the incidence of complications, which are sometimes lethal. Maintaining an EVD through careful administration of fibrinolytic drugs helps to avoid occlusion, but choosing which patient/EVD should be subjected to intervention requires a ‘customized approach’. uPA administration must be preceded by a careful risk assessment and must be performed in a sterile environment with continuous ICP monitoring. Complications such as bleeding and infection are rare, but the choice of whether to administer IVF must always be personalized for the patient.

## Figures and Tables

**Figure 1 jpm-15-00264-f001:**
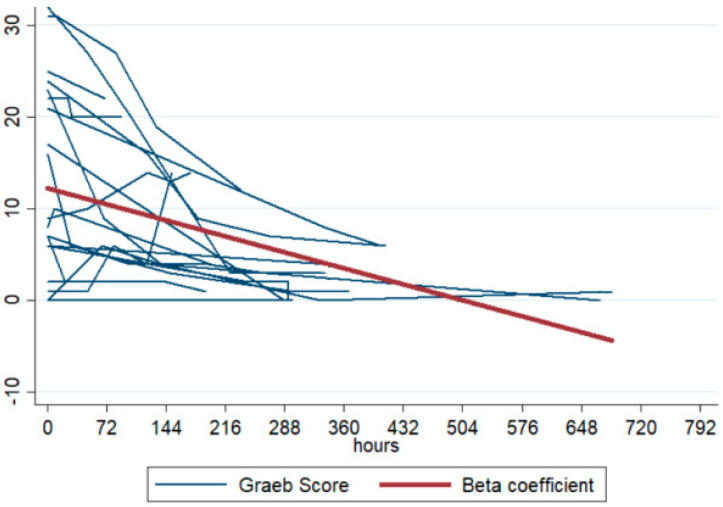
β coefficient, hours versus Graeb score.

**Table 1 jpm-15-00264-t001:** Demographic and anthropometric variables, diagnoses, and comorbidities. The patient aged 57 who was diagnosed with thalamic ICH + IVH experienced an episode of asymptomatic bleeding following IVF.

Age	Sex	Weight	Height	BMI	Comorbidities	ICU Admission Diagnosis	GCS at Onset	IVH y/n	Hydrocephalus y/n
22	M	58	165	21.3	moderate COPD	cerebellar left ICH by left ACM aneurysm rupture	13	y	y
25	F	50	160	19.5	systemic hypertension, heart failure NYHA 4, AF, COPD, TEP	ICH + IVH	7	y	y
38	F	80	150	35.5	multinodular goiter	pancisternal SAH by subtentorial AVM rupture	3	y	y
44	F	133	165	48.8	none	pancisternal SAH by ACoP aneurysm rupture	4	y	y
46	M	105	180	32.4	systemic hypertension, hepato-renal poly-cistosis, SAH by PICA	suspected mycotic cerebral infection + IVH	3	n	y
51	F	60	165	22.0	obesity, systemic hypertension, previous TEP, hyperuricemia	IVH post trans-sphenoidal craniopharyngioma removal	12	y	y
51	F	58	165	21.3	none	ICH + IVH by mesencephalic AVM rupture	3	y	y
53	F	55	160	21.4	obesity, Lobstein syndrome	SAH by carotid siphon aneurysm rupture	8	y	y
57	F	65	160	25.4	obesity, systemic hypertension, type 2 diabetes mellitus	thalamic ICH + IVH	7	y	y
60	M	60	172	20.3	systemic hypertension	SAH by PCA 1 dissection	12	y	y
61	M	69	180	21.3	HIV, HCV, previous TIA	meningo-ventriculitis associated with drug abuse	7	n	n
63	M	74	180	22.8	obesity, hypopituitarism	ICH + IVH in Moya-Moya syndrome	5	y	y
63	M	85	170	29.4	obesity, COPD, cardiac failure, CKD	PRES	8	n	y
64	M	65	147	30.1	systemic hypertension	meningo-ventriculitis after craniopharyngioma removal	13	y	y
64	F	75	160	29.3	AF, systemic hypertension, obesity	cerebellar ICH + IVH	15	y	y
66	M	110	185	32.1	severe psychosis, previous removal of pituitary adenoma, panhypopituitarism	obstructive hydrocephalus with suspected meningitis	8	n	y
70	M	90	180	28.0	previous AF subjected to ablation	SAH	14	y	y
74	M	92	175	30.0	congenital mono-kidney	SAH by AcoA aneurysm rupture + IVH	3	y	y
76	F	85	170	29.4	none	cerebellar ICH + IVH	4	y	y
79	F	60	160	23.4	systemic hypertension, previous TIA	SAH by BA dissection + IVH	3	y	y

Body mass index (BMI), chronic obstructive pulmonary disease (COPD), intracranial hemorrhage (ICH), middle cerebral artery (ACM), New York Heart Association (NYHA), atrial fibrillation (AF), pulmonary trombo-embolism (TEP), intraventricular hemorrhage (IVH), subarachnoid hemorrhage (SAH), artero-venous malformation (AVM), posterior communicating artery (ACoP), posterior inferior cerebellar artery (PICA), posterior cerebral artery (PCA), positivity for human immunodeficiency virus / hepatitis C virus (HIV/HCV), transient ischemic attack (TIA), chronic kidney disease (CKD), posterior reversible encephalopathy syndrome (PRES), anterior communicating artery (AcoA), basilar artery (BA).

## Data Availability

The data presented in this study are available on request from the corresponding author.

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
