# Peer review of "A Personalized Approach to Maintaining Brain Drainage: A Case Series with a Technical Note"

_jpm, 2025, doi:10.3390/jpm15070264_

Round 1

Reviewer 1 Report

Comments and Suggestions for Authors

The authors present a very pertinent study analyzing 20 patients with external ventricular drain (EVD) receiving intrathecal urokinase-type plasminogen activator (uPA) for intraventricular fibrinolysis (IVF). 95% of the patients presented with hydrocephalus and 80% with intraventricular hemorrhage; uPA dosages varied (25,000–100,000 IU), with an average of 3.9 doses per patient. IVF effectively maintained EVD patency in 95% of cases. Asymptomatic bleeding occurred in one patient (5%) and post-treatment infections developed in four patients (20%). Faster clot resolution with early uPA administration was achieved in Graeb score analysis. A higher initial Graeb score correlated with increased total uPA load but not with mortality or discharge outcomes.

The study design is well done and overall aim is of interest. The results are very promising and the study has clinical impact on the postoperative outcome of the patients.

There are major revision requests.

Specific comments are enumerated below:

  • In the methods section, the authors should state on inclusion and exclusion criteria for patients receiving uPA for IVF.
  • In the methods section, the authors should provide more information for the timing of uPA administration in patients with SAH: Did the patients receive uPA after clipping or coiling of the aneurysm or at initial diagnosis before aneurysm occlusion? The therapeutic modality should also be included into the table.
  • In the methods section, the authors should include a short description of the Graeb score and how the Graeb score is calculated
  • It should be evaluated if Table 1 and Table 2 could be merged as some of the patient comorbidities (obesity) of Table 1 are also included in Table 2 and some parameters do not seem to be relevant for the study.
  • Figure 3 does not provide relevant information for the reader; therefore, it could be omitted. Instead, it would be interesting to see the CT scans of the patient with the asymptomatic bleeding
  • For the patient outcome, it would be interesting for the reader to know, which patient presented the asymptomatic bleeding and the post-treatment infections. This information should be included to table 2 or in a new table.
  • The description of uPA with “preparatory and operational phase” as well as Figure 4a-c should be included to the supplementary files. Instead, the discussion section could be enriched.
  • In the results section, the authors state that “8 of the patients observed (40%) received a diagnosis of Central Nervous System (CNS) infection during hospitalization in an intensive care setting (clinical signs and laboratory confirmation with positive culture tests on CSF). Of these, 4 had a CNS infection diagnosis that preceded EVD placement or uPA administration, while 4 (20% of all the patients treated) occurred following treatment with uPA.” In the discussion section, however, the authors describe “However, cases of infection appear to be more frequent: 10%.” Could the authors please correct the rate of infection (20 vs. 10%)? Could the authors also state on the high infection rate compared to the rate in the CLEAR-III trial (4%)?
  • In the discussion section, the authors should consider about the fact that CNS infection appeared in 2 patients out of 4 patients after 21 days. They should discuss about a potential cut-off day for EVD duration.
  • In the discussion section the authors describe that in this study “… the clinicians, [chose] in a ‘customised approach’, […] to treat with intraventricular administration of uPA for impending catheter occlusion.” Also the title describes "A personlized approach [...]." However, the selection criteria of the patients remains unclear. Please clarify on that in the methods and discussion section.

Author Response

In the methods section, the authors should state on inclusion and exclusion criteria for patients receiving uPA for IVF. Response: Thank you. I've added statements in the methods section about inclusion criteria for patients receiving uPA.

In the methods section, the authors should provide more information for the timing of uPA administration in patients with SAH: Did the patients receive uPA after clipping or coiling of the aneurysm or at initial diagnosis before aneurysm occlusion? The therapeutic modality should also be included into the table. Response: Thank you. I've added informations about the timing of uPA administration in the methods section.

In the methods section, the authors should include a short description of the Graeb score and how the Graeb score is calculated. Response: Thank you. I've added statements in the methods section about calculation of Graeb score.

It should be evaluated if Table 1 and Table 2 could be merged as some of the patient comorbidities (obesity) of Table 1 are also included in Table 2 and some parameters do not seem to be relevant for the study. Response: Thank you. I've merged Table 1 and Table 2.

Figure 3 does not provide relevant information for the reader; therefore, it could be omitted. Instead, it would be interesting to see the CT scans of the patient with the asymptomatic bleeding. Response. Thank you. I've removed Figure 3, I've added CT scan of the asymptomatic patient in the supplemental materials.

For the patient outcome, it would be interesting for the reader to know, which patient presented the asymptomatic bleeding and the post-treatment infections. This information should be included to table 2 or in a new table. Response: Thank you. I've added this information in Table 1.

The description of uPA with “preparatory and operational phase” as well as Figure 4a-c should be included to the supplementary files. Instead, the discussion section could be enriched. Response: Thank you. I've added the "preparatory and operational phase" to supplementary files.

In the results section, the authors state that “8 of the patients observed (40%) received a diagnosis of Central Nervous System (CNS) infection during hospitalization in an intensive care setting (clinical signs and laboratory confirmation with positive culture tests on CSF). Of these, 4 had a CNS infection diagnosis that preceded EVD placement or uPA administration, while 4 (20% of all the patients treated) occurred following treatment with uPA.” In the discussion section, however, the authors describe “However, cases of infection appear to be more frequent: 10%.” Could the authors please correct the rate of infection (20 vs. 10%)? Could the authors also state on the high infection rate compared to the rate in the CLEAR-III trial (4%)? Response: Thank you. I've corrected the infection rate. I've also added a statement on the infection rate observed.

In the discussion section, the authors should consider about the fact that CNS infection appeared in 2 patients out of 4 patients after 21 days. They should discuss about a potential cut-off day for EVD duration. Response. Thank you: I've added a statement in the discussion section.

In the discussion section the authors describe that in this study “… the clinicians, [chose] in a ‘customised approach’, […] to treat with intraventricular administration of uPA for impending catheter occlusion.” Also the title describes "A personlized approach [...]." However, the selection criteria of the patients remains unclear. Please clarify on that in the methods and discussion section. Response: Thank you. I've added selection criteria in discussion section.

Reviewer 2 Report

Comments and Suggestions for Authors

GENERAL

The entire manuscript needs to be edited for English language and punctuation. 

INTRODUCTION

Please un-capitalize Acute Brain Injuries and Intracranial Pressure, etc. I would truncate the introduction by a few sentences it is a bit too long and wordy. 

METHODS

I recommend expanding the statistical methods section - model assumptions, handling of missing data, CI interpretation.

Include a multivariate or stratified infection risk analysis (e.g., based on EVD duration).

DISCUSSION

Include practical guidance on selecting uPA dose (25,000 vs. 50,000 IU) based on Graeb score or clot burden.

The limitations and future directions is absent from the discussion. 

CONCLUSIONS

Its important to keep the conclusions in line with data presented. Broader conclusions regarding comparative efficacy, infection risk, and long-term outcomes should be more cautiously stated and explicitly limited by the study’s retrospective design, small sample size, and lack of control.

Comments on the Quality of English Language

Poor- The entire manuscript needs to be edited for English language and punctuation. 

Author Response

Comment 1: The entire manuscript needs to be edited for English language and punctuation. Response: Thank you. I've revised the entire document.

Comment 2: Please un-capitalize Acute Brain Injuries and Intracranial Pressure, etc. I would truncate the introduction by a few sentences it is a bit too long and wordy. Response: Thank you. I've un-capitalized the words you suggested. I've reduced the lenght of the introduction.

Comment 3: I recommend expanding the statistical methods section - model assumptions, handling of missing data, CI interpretation. Response: We added clarification in the Methods section regarding model assumptions and the interpretation of the 95% confidence intervals: “The overall error distribution in linear mixed-effects models is assumed to be Gaussian. Additionally, the model can accommodate heteroskedasticity and within-group correlation structures by specifying appropriate covariance structures for the random effects and residuals. This flexibility allows for accurate modeling of longitudinal data where repeated measures are nested within subjects.
The 95% confidence intervals (95% CI) presented for the β coefficients represent the range within which the true population parameter is expected to lie with 95% confidence, assuming the model is correctly specified and its assumptions are met.”

Comment 4: Include a multivariate or stratified infection risk analysis (e.g., based on EVD duration). Response: We agree that such an analysis would be valuable; however, due to the limited number of infection events in our cohort (n=10), conducting a reliable multivariable model was not statistically feasible. We clarified our approach in the Methods section: “The associations between infection risk and clinical variables, such as EVD duration, GOS-E at discharge and total uPA load, were evaluated using the Kruskal-Wallis test.”

Comment 5: Include practical guidance on selecting uPA dose (25,000 vs. 50,000 IU) based on Graeb score or clot burden. Response: Thank you. I've added a sentence in the Discussion section about the choice of the dose of uPA.

Comment 6: The limitations and future directions is absent from the discussion. Response: Tahnk you. I've added limitations and indications for future studies at the end of the Discussion section.

Round 2

Reviewer 1 Report

Comments and Suggestions for Authors

I would like to thank the Authors for carefully considering my suggestions and for their thorough revisions. The manuscript has been significantly improved, addressing all the key points raised in my initial review. I am satisfied with the modifications made and have no further comments or suggestions.

Reviewer 2 Report

Comments and Suggestions for Authors

The authors have reviewed my comments and answered the questions appropriately while revising the text.